# Advancing Multi-grained Alignment for Contrastive Language-Audio Pre-training

## ABSTRACT

Recent advances have been witnessed in audio-language joint learning, such as CLAP, that shows much success in multi-modal understanding tasks. These models usually aggregate uni-modal local representations, namely frame or word features, into global ones, on which the contrastive loss is employed to reach coarse-grained cross-modal alignment. However, frame-level correspondence with texts may be ignored by the above paradigm, making it ill-posed on explainability and fine-grained text-audio challenges (e.g., text-to-audio grounding) which may also undermine performances on coarse-grained tasks. In this work, we aim to improve both coarse- and fine-grained audio-language alignment in large-scale contrastive pre-training. To unify the granularity and latent distribution of two modalities, a shared codebook is adopted to represent multi-modal global features with common bases, and each internal codeword is regularized to encode modality-shared semantics, bridging the gap between frame and word features. Based on the above framework, a locality-aware block is involved to purify local patterns, and a hard-negative guided loss is devised to boost alignment effects. Extensive experiments on eleven zero-shot coarse- and fine-grained evaluation protocols suggest that our model not only surpasses the baseline CLAP significantly but also yields superior or competitive results compared to current SOTA works. The code and model will be released upon paper acceptance.

## CCS CONCEPTS

• **Information systems → Speech / audio search**.

## KEYWORDS

Contrastive language-audio pre-training, Zero-shot inference, Audio-text retrieval, Fine-grained interaction

## 1 INTRODUCTION

Sound conveys a lot of information in our daily lives. With the advance of learning theories and data collections [11], large-scale pre-trained models, such as PANNs [21] and AST [13], have witnessed extraordinary achievements on sound-related challenges, such as sound classification [38] and sound event detection [26]. Despite such success, these methods still require downstream tuning to adapt to novel scenarios and cannot facilitate tasks related to natural language, e.g., retrieve or generate audio clips [28, 51, 52] according

*ACM MM, 2024, Melbourne, Australia*

© 2024 Copyright held by the owner/author(s). Publication rights licensed to ACM.
ACM ISBN 978-x-xxxx-xxxx-x/YY/MM
https://doi.org/10.1145/nnnnnnn.nnnnnnn

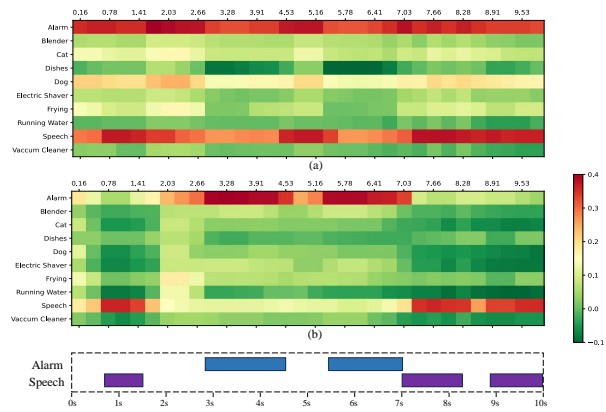

**Figure 1: The cosine similarity between frame features of (a) CLAP, (b) our MGA-CLAP and text features of ten sound classes. The sample audio contains alarm (2.8s- 4.5s, 5.4s-7.0s) and speech (0.6s-1.5s, 7.0s-8.3s, 8.9s-10.0s).**

to human instructions. Alternatively, Contrastive Language-Audio Pre-training (CLAP) [9] is introduced to learn general and transferable representations by associating audio samples with corresponding captions. Consequently, an aligned feature space is built, making it versatile for several tasks, such as zero-shot audio classification and retrieval, by simply computing the cosine similarity between encoded audio and textual features of sound classes [25].

However, during empirical practices, we notice that current CLAP models lack the capability of capturing the fine-grained alignment like the relationship between acoustic events and textual meanings. An example of this phenomenon is depicted in Figure 1 (a). As seen, although the two kinds of events, namely alarm and speech, are successfully recognized by the original CLAP, the similarity between frame representations and textual sound representations shows much inconsistency with the real temporal locations of sound events. For instance, the sound "alarm" is recorded at 2.8s-4.5s and 5.4s-7.0s, but the corresponding frame-level similarity is high over the whole clip. This may undermine the model's explainability and lead to undesirable results on fine-grained cross-modal understanding tasks, including zero-shot sound event detection and text-to-audio grounding [53]. Moreover, poor performance can also be observed in certain cases when conducting coarse-grained tasks like zero-shot audio tagging and retrieval, since local patterns and temporal information are potentially ignored by the vanilla CLAP paradigm. We attribute the above problem to the lack of interaction between frame and word features during CLAP training, as current CLAP methods reach the cross-modal alignment via solely the similarity of the global features of each modality.

To mitigate the research gap, we propose to adopt a modality-shared codebook to encourage the multi-modal features to interact on a finer granularity. The codebook consists of several learnable

codewords, and the weighted summation of them will be utilized to represent the global features of each modality, so that they are naturally restricted in the same feature space, making it easier to learn the alignment. To encode cross-modal shared semantic concepts (e.g., sound events) into each codeword, we further revise the traditional working scheme of the codebook to compute the aggregation weights. Practically, we define the affinity scores between a clip (or a caption) and each codeword as the maximum cosine similarity between its frame features (or word features) and the specific codeword. Then, the global feature can be represented with a small number of codewords by applying sparse constraints on the affinity scores to avoid noisy activation before using them as aggregation weights. Through optimizing the contrastive loss, not only the paired global features can be well-aligned, but also the frame features of an acoustic event (e.g., "alarm") and the word features of the corresponding caption can activate the same, small set of codewords, thereby implicitly building a connection between fine level multi-modal features. Moreover, we also notice that local acoustic patterns may be destroyed by the vanilla transformer block and devise a novel locality-aware block to ensure high-quality frame features for codewords aggregation. Finally, a hard-negative guided contrastive loss is reformulated to mine more discriminative representations in order to build a better-aligned global latent space. Equipped with these techniques, our MGA-CLAP reaches a better fine-grained alignment than the original CLAP without losing its natural coarse-grained alignment, as shown in Figure 1 (b).

We conduct extensive experiments on both coarse- and fine-grained audio-text tasks. As for the fine-grained ones, MGA-CLAP surpasses the original CLAP to a large extent. Specifically, using WavCaps [30] as the main pre-training dataset, MGA-CLAP achieves 26.4%/10.1% PSDS1 on zero-shot DESED [41]/AudioSet-Strong [15] sound event detection tasks, which is 13.3%/6.7% higher than its baseline CLAP. While for coarse-grained retrieval and tagging tasks, our method also demonstrates noticeable improvements over CLAP and shows better performance on most evaluation protocols compared to previous SOTA works which generally require much more training resources. Besides, several ablation studies are performed to reveal the effects of each component elaborately. Finally, we also visualize the semantic meanings of specific codewords to show their roles in linking different modalities.

## 2 RELATED WORK

### 2.1 Contrastive Language-Audio Pre-training

By pre-training on 400M image-text pairs, CLIP [33] demonstrates superior performance and transferability on cross-modal vision problems, such as zero-shot image retrieval and classification. Several works, including AudioCLIP [14] and Wav2CLIP [47], try to leverage visual modality as a bridge to connect text and audio representations, achieving promising results on zero-shot audio tagging tasks. With the collection of large-scale audio caption datasets, namely AudioCaps [19], Clotho [7] and WavText5K [6], a lot of works explore contrastive language-audio pre-training without involving the visual modality. MS-CLAP [9] first obtains aligned text and audio encoders on a combination of off-the-shelf audio-text datasets. However, due to the limits of data size, its performance is sub-optimal. A few researchers then turn to expand the

scale of audio-text datasets. LAION-Audio-630K [48] and WavCaps [30], collected and annotated by human professionals and Chat-GPT respectively, are shown to be more effective for pre-training. Besides, BLAT [55] proposes to utilize a well-trained model together with audio tags to automatically generate audio captions for contrastive pre-training while Cacophony [59] combines an audio caption model and Large Language Models (LLMs) to expand the data size to 4M and explore training strategies on such large-scale dataset. Moreover, the intrinsic shortcomings of CLAP are also studied. ACBA [46] and CompA [12] enhance CLAP's compositional reasoning ability while FLAP [57] devises masking strategies to improve both the training efficiency and model performance. By contrast, we notice the unsatisfactory fine-grained alignment of CLAP and aim to discover both fine-grained and coarse-grained correspondence solely from audio-text pairs.

### 2.2 Audio Feature Learning with Codebook

Codebook is the key design in vector quantization [42], which is widely adopted for both understanding [1] and generation [36] tasks. During quantization, encoder features will be substituted by their nearest-neighbor codewords in the codebook, before being utilized to reconstruct original features by the decoder. Finally, by querying the learned codebook, a continuous space can then be transformed into finite discrete tokens. Following this way, modern neural audio codec models [49, 58] learn to convert the raw waveform into several codewords, paving the way for efficient audio compression [5] and auto-regressive audio generation [44]. Besides, BEATs [4], the state-of-the-art self-supervised learning approach, also employs an acoustic tokenizer to quantize spectrograms into codewords for mask prediction, which demonstrates better performance compared to reconstruction methods, namely AudioMAE [17]. Different from the above, we leverage the codebook to accommodate both text and audio hidden representations instead of single-modality raw signals, which explicitly constructs a shared multi-modal feature space for coarse-grained alignment. Moreover, we dedicatedly redesign the computational rules of the codebook so that it can help discover the fine-grained correspondence.

### 2.3 Learning Frame-level Correspondence from Weak or Caption Supervision

Frame-wise labeling is extremely laborious for audio tasks, hence learning from partially labeled data (e.g., weak labels or audio captions) becomes a promising remedy. Weakly supervised sound event detection [22, 27] aims to recognize the sound event boundary under weak supervision, where only the clip-level annotations are provided but the exact timestamps are inaccessible. However, it solely maps acoustic features to a closed label set, which limits its applications in open-world scenarios. By contrast, learning from audio captions addresses the aforementioned problem by associating frame features with general language descriptions. But it is more challenging due to the intrinsic modality gap. Besides, the noisy information (non-sound words) contained in the captions also increases the difficulty. UACA [50] first learns relationships between sound events and textual phrases from audio captions by aggregating frame-word similarity matrix to clip-caption similarity, while WSTAG [54] improves it by leveraging max-mean instead of

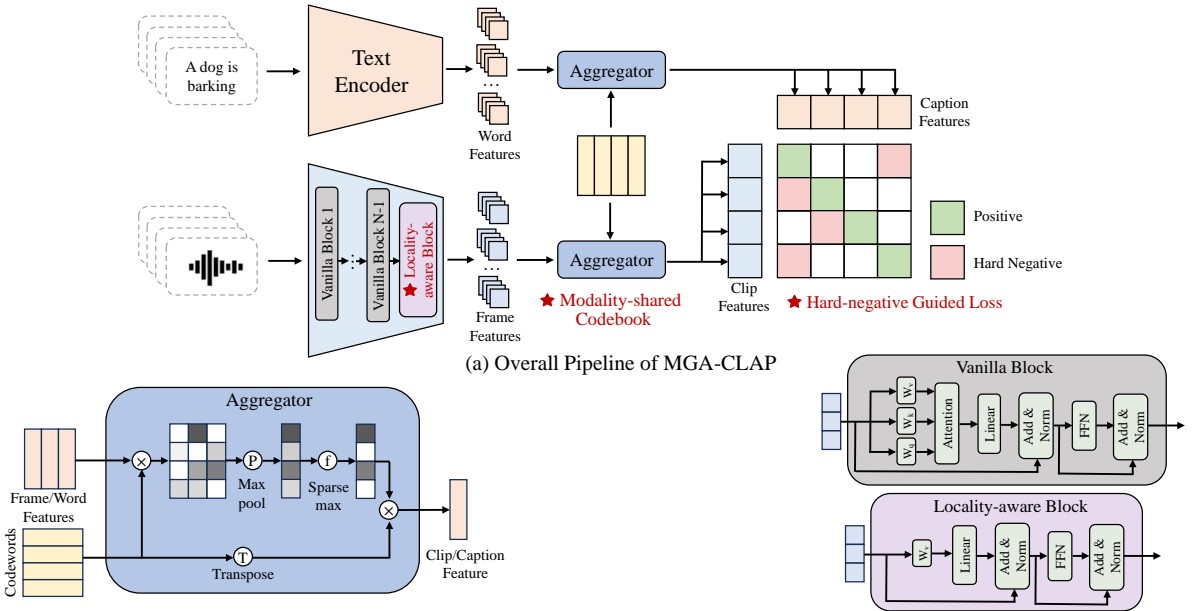

(a) Overall Pipeline of MGA-CLAP

(b) Illustration of the Modality-shared Codebook

(c) Comparison between Vanilla and Locality-aware Block

Figure 2: (a) shows the overall pipeline of our MGA-CLAP. (b) illustrates the aggregation mechanism of the codebook. (c) demonstrates the key difference between the proposed locality-aware block and vanilla transformer block.

mean-mean pooling. However, these works depend on exhaustive score matching while ignoring complex frame-word interaction, leading to suboptimal fine-grained alignment when scaling to a much larger pre-training dataset. In this work, we propose a novel solution to model the frame-word correspondence, demonstrating better performance and scalability than [50, 54].

## 3 METHODOLOGY

### 3.1 Overview

An overview of our MGA-CLAP is shown in Figure 2 (a). As illustrated before, we introduce a novel modality-shared codebook, which aggregates frame- and word-level features with shared codewords. Then, in order to refine the frame-wise features, the locality-aware block is involved to better capture local patterns. Finally, the CLAP loss is reformulated to emphasize indistinguishable audio-text pairs for contrastive optimization. In the following subsections, we will detail the above three core designs.

### 3.2 Modality-shared Codebook

*3.2.1 Multi-modal Representations in CLAP.* CLAP employs a bi-encoder architecture to learn the aligned feature space for both modalities. Specifically, assume that we have a batch of audio-text pairs $\{(x_i, y_i)\}_{i=1}^{B}$, where $x_i$, $y_i$ represent the $i$ th audio clip and its caption, and $B$ is the batch size. CLAP audio encoder $f$ takes $x_i$ as input before generating frame representations $P_i = f(x_i) \in \mathbb{R}^{T \times D}$ while the text encoder $g$ outputs word-level features $Q_i = g(y_i) \in \mathbb{R}^{N \times D}$ according to $y_i$, where $N$, $F$ and $D$ is the number of frames, words and feature dimensions, respectively. Then, to obtain the global clip- and caption-level feature, an aggregator $h^{(a)} : \mathbb{R}^{T \times D} \to \mathbb{R}^{D}$ is required to map $f(x_i)$ to $\hat{p}_i$, and $h^{(t)}$ works

similarly to aggregate $g(y_i)$ to $\hat{q}_i$. Finally, the symmetric contrastive loss is optimized to pull together the global features of paired audios and texts while pushing away unpaired ones in the latent space,

$$\mathcal{L}_{\text{CLAP}} = -\sum_{i=1}^{B} log \frac{e^{<\hat{p}_i, \hat{q}_i>/\tau}}{\sum_{j=1}^{B} e^{<\hat{p}_i, \hat{q}_j>/\tau}} - \sum_{i=1}^{B} log \frac{e^{<\hat{q}_i, \hat{p}_i>/\tau}}{\sum_{j=1}^{B} e^{<\hat{q}_i, \hat{p}_j>/\tau}} \quad (1)$$

where $< \cdot, \cdot >$ is the inner product function and $\tau$ is a scaling factor.

In the above CLAP paradigm, $h^{(a)}$ and $h^{(t)}$ are instantiated by mean pooling or attention pooling, suggesting that the global features are essentially weighted sum of two bases: the audio frames and language tokens. However, due to the modality gap, the two bases may exhibit different granularities and semantics thus distributed in distinct hidden spaces, making it challenging to learn the coarse-grained alignment. Moreover, the frame and word representations are separately aggregated to the global features without additional interactions, which may increase the difficulty of discovering more granular correspondence (e.g., the frame-to-word, frame-to-phrase alignment) since only the coarse-level supervision is accessible in audio-text pairs.

*3.2.2 Modality-shared Codebook as the Aggregator.* To seek a common multi-modal hidden space, we introduce a novel modality-shared codebook as the feature aggregator. By this means, the global audio feature $\tilde{p}_i$ and text feature $\tilde{q}_i$ are represented with the same set of $M$ learnable codewords as $\tilde{p}_i = \sum_{k=1}^{M} w_{i,k}^{(a)} z_k$ and $\tilde{q}_i = \sum_{k=1}^{M} w_{i,k}^{(t)} z_k$, where $\{z_k | z_k \in \mathbb{R}^{D}, k = 1, 2, \cdots M\}$ are the mentioned codewords, and $w_{i,k}^{(a)}$, $w_{i,k}^{(t)}$ are the corresponding aggregation weights of $z_k$ for clip $x_i$ and caption $y_i$, respectively. To capture rich local semantics during aggregation, we specially devise the pipeline to calculate $w_{i,:}^{(a)}$ and $w_{i,:}^{(t)}$ as shown in Figure 2 (b).

Mathematically, given the extracted frame-wise features $P_i$ of clip $x_i$, we define the affinity score $s_{i,k}^{(a)}$ between $x_i$ and $z_k$ as,

$$s_{i,k}^{(a)} = \max_j < P_{i,j}, z_k > /\eta \tag{2}$$

where $P_{i,j} \in \mathbb{R}^D$ is the $j$ th frame feature of $P_i$ and $\eta$ is a scaling term. Notably, adopting max pooling instead of mean pooling may uncover momentary sounds even if they only last one frame, thereby guaranteeing semantic integrity during aggregation.

The affinity scores $s_{i,:}^{(a)}$ are then normalized by the Sparsemax [29] function, which works similarly to Softmax but encourages most of the elements in the probability distribution to be 0.

$$w_{i,:}^{(a)} = \text{Sparsemax}(s_{i,:}^{(a)}) \tag{3}$$

By the sparse constraints, $\tilde{p}_i$ can be represented by only a few codewords, which helps eliminate the noisy activation and enhance the interpretability. Similarly, the global text feature $\tilde{q}_i$ can also be constructed via the above way.

Finally, we provide an intuitive view of how the proposed paradigm reaches fine-grained cross-modal alignment. Under the supervision of contrastive loss, the similarity of paired samples $< \tilde{p}_i, \tilde{q}_i >$ is supposed to be maximized. However, due to the sparse regularization, the model may have to resort to the same, small set of codewords to represent the audio $x_i$ and text $y_i$ to increase $< \tilde{p}_i, \tilde{q}_i >$. Let $k^*$ be one of the activated codewords. It then acts as prior targets, which requires the encoders to refine frame (or word) representations to maximize $s_{i,k^*}^{(a)}$ and $s_{i,k^*}^{(t)}$. As a result, the corresponding frame and word features are then attracted to the same anchor $z_{k^*}$ which contains semantic information of specific sound classes, thereby bridging the gap between multi-modal local features.

### 3.3 Locality-aware Encoder Block

Obtaining meaningful local representations is crucial, otherwise, some codewords may be activated by mistake during feature aggregation. Recall that in the vanilla CLAP audio encoder, the outputs of the last transformer encoder block will be decoupled to produce the final frame-wise features. Its general architecture can be found in the upper of Figure 2 (c), which first employs self-attention to consider global contexts. Specifically, let $U = \{u_l | u_l \in \mathbb{R}^d\}_{l=1}^T$ be the input sequence of the block, the query, key, value matrices $Q, K, V$ are first calculated by separate linear projections $W_q, W_k, W_v$,

$$Q = W_q U, K = W_k U, V = W_v U \tag{4}$$

Then, to compute output feature $u_l'$ for each frame $l$, the q-k attention is applied as follows,

$$attn\_score_{l,:} = \text{Softmax}(< q_l, K > /\sqrt{d}) \tag{5}$$

$$u_l' = \sum_j attn\_score_{l,j} \cdot v_j \tag{6}$$

where $q_l$ and $v_j$ is the $l$ th and $j$ th vector of $Q$ and $V$. In this way, information from other frames $v_j$ can be injected into the current frame $l$ by referring to the q-k similarity.

However, according to the mechanism of self-attention [43], we argue that $v_l$ computed at each location $l$ already captures rich local semantics. By contrast, obtaining a comprehensive view by attention aggregation may impurify the local patterns, which may be a

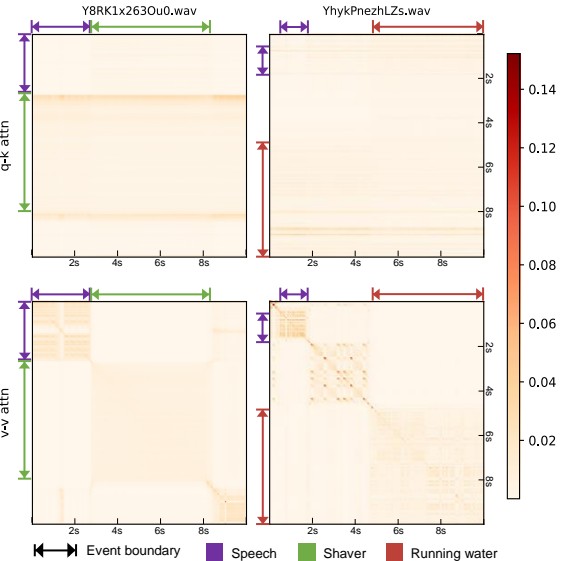

**Figure 3: The q-k (the 1st row) and v-v similarities (the 2nd row) along time axis of two sample audios. The scores are detached from the last encoder block of the original CLAP. And the sound boundary is marked with double sided arrow.**

negative for fine-grained alignment. Figure 3 gives two examples to support our hypothesis. As seen, the v-v similarities (computed by Softmax($< v_l, V > /\sqrt{d}$), which is similar to Equation (5)) of the last block are high within the same sound event while low among different events, meaning that acoustically dissimilar frames may exhibit distinct $v$ for finer-level discrimination. In comparison, the q-k similarities show inconsistency with the event boundary.

Inspired by the above, we design the locality-aware block as shown in Figure 2 (c), which simply removes the q-k attention and directly leverages the projected value matrix $V$ as the output feature sequence $U'$ with other components unchanged. Besides, we only replace the last block of the audio branch with the locality-aware block, the reasons are: (1) the transformer receptive fields become much more global midway through the network as [35] suggest; (2) since the audio encoder is pre-trained on AudioSet to learn general acoustic patterns (a widely-adopted setting of previous CLAP variants), replacing the last one can retain most prior knowledge.

### 3.4 Hard Negative Guided Contrastive Loss

Contrastive learning can benefit a lot from in-batch hard negative samples [37]. For vision-language tasks, several works [23, 45, 56] tend to resample or manually craft hard negative instances to improve alignment effects, which generally involves more training costs. In this work, we devise a simple re-weighting approach to force the modal to pay more attention to hard negative samples during optimization. The loss function is reformulated as follows,

$$\mathcal{L}_{\text{HN\_CLAP}} = -\sum_{i=1}^B \log \frac{e^{<\tilde{p}_i, \tilde{q}_i>/\tau}}{e^{<\tilde{p}_i, \tilde{q}_i>/\tau} + \sum_{j, j\neq i} \alpha_{i,j} e^{<\tilde{p}_i, \tilde{q}_j>/\tau}}$$
$$-\sum_{i=1}^B \log \frac{e^{<\tilde{q}_i, \tilde{p}_i>/\tau}}{e^{<\tilde{q}_i, \tilde{p}_i>/\tau} + \sum_{j, j\neq i} \beta_{i,j} e^{<\tilde{q}_i, \tilde{p}_j>/\tau}} \tag{7}$$

where $\alpha_{i,j}$, $\beta_{i,j}$ is the audio-to-text and text-to-audio difficulty scores for unpaired samples, they are designed so that hard negative pairs (with higher similarity compared to the average score) are emphasized in loss computation, and easier pairs are neglected. As a result, the model will be forced to learn a more discriminative feature space to distinguish confusable pairs for multi-grained alignment. The formula is written as,

$$\alpha_{i,j} = \frac{Be^{\gamma<\tilde{p}_i,\tilde{q}_j>/\tau}}{\sum_k e^{\gamma<\tilde{p}_i,\tilde{q}_k>/\tau}}, \quad \beta_{i,j} = \frac{Be^{\gamma<\tilde{q}_i,\tilde{p}_j>/\tau}}{\sum_k e^{\gamma<\tilde{q}_i,\tilde{p}_k>/\tau}} \quad (8)$$

where $\gamma$ is a scaling ratio, the larger it is, the more importance we attach to the hard negative samples as the distribution of $\alpha_{i,:}$ and $\beta_{i,:}$ can be sharper.

## 4 EXPERIMENTAL SETUP

### 4.1 Pre-training

**Dataset.** We merge WavCaps, the training set of AudioCaps and Clotho for pre-training, including about 450K audio-text pairs.

**Architecture.** We employ the pre-trained BERT [18] base model as the text encoder which contains 110M parameters. While for the audio encoder, to examine the scalability of the proposed method, we adopt a patch-wise model HTS-AT (27M) [3] and a frame-wise AST (86M) [24], all of them are trained on the AudioSet by previous works and we directly use the checkpoints. Besides, a two-layer MLP is appended after the encoder, projecting the multi-modal features into the same dimension $D = 1024$.

**Implementation Details.** We train our model for 15 epochs with a batch size of 128 and a learning rate of 5e-5 using the Adam optimizer. The hyper-parameter $\tau$ is learnable with an initial value of 0.07 and $\gamma$ and $M$ are fixed to 0.15 and 4096 empirically. Besides, all the audio clips and captions are randomly cropped or padded to 10 seconds and 30 words to guarantee the fixed-sized length. We also resample the waveform to 32KHz and 16KHz for HTS-AT and AST following the original works. During training, audio clips with similar durations are grouped within a batch for training efficiency. Finally, model checkpoints are selected based on their performance on validation sets after each epoch and the final model performances are evaluated on corresponding test sets.

### 4.2 Downstream Evaluation

To comprehensively evaluate the model performance, we conduct experiments on several coarse-grained tasks (including audio retrieval, audio classification, and audio tagging) and fine-grained tasks (including sound event detection and text-to-audio grounding). Note that for each specific task, we would pre-train a new model from scratch with a newly constructed dataset excluding all the overlapped samples in the downstream evaluation, meaning that the zero-shot inference is performed. Specifically, for tasks other than retrieval, we directly use sound class names as the input of the text encoder, which avoids heavy prompt engineering. We will report the averaged metric of 3 different runs and the detailed evaluation protocols are provided in Table 1. For single-label and multi-label classification tasks, Acc and mAP are widely adopted metrics. For retrieval tasks, R@k is 1 if the positive item appears in the top k retrieved items for a query [20]. And for detection and grounding tasks, PSDS1 is more sensitive to the precise localization

**Table 1: Evaluation datasets and metrics for each task.**

| Task | Datasets | metrics |
|---|---|---|
| audio retrieval | AudioCaps (AC), Clotho | R@1, R@5 |
| audio classification | ESC-50 [32], UrbanSound8K (US8K) [39], VGGSound [2] | Acc |
| audio tagging | FSD50K [10], AudioSet (AS) | mAP |
| sound event detection | DESED, UrbanSED [40], AudioSet-Strong (AS-S) | PSDS1, PSDS2 |
| text-to-audio grounding | TAG [53] | PSDSm |

**Table 2: Performance comparison on zero-shot audio-text retrieval tasks. Models marked with ⁺/* are based on the HTS-AT/AST backbone.**

| Model | AC | | | | Clotho | | | |
|---|---|---|---|---|---|---|---|---|
| | Text2Audio | | Audio2Text | | Text2Audio | | Audio2Text | |
| | R@1 | R@5 | R@1 | R@5 | R@1 | R@5 | R@1 | R@5 |
| FLAP (fusion) | 41.5 | 75.5 | 53.0 | 84.1 | 20.3 | **46.5** | 25.5 | 53.4 |
| Cacophony | 41.0 | 75.3 | **55.3** | 83.6 | 20.2 | 45.9 | **26.5** | **54.1** |
| CLAP⁺ | 39.7 | 74.5 | 51.9 | 82.1 | 19.5 | 45.2 | 23.4 | 50.7 |
| MGA-CLAP⁺ | 41.8 | **76.1** | 54.4 | 83.6 | 20.4 | 46.0 | 25.3 | 51.2 |
| CLAP* | 40.1 | 74.0 | 51.8 | 82.4 | 18.5 | 43.3 | 23.9 | 51.6 |
| MGA-CLAP* | **42.2** | 74.9 | 53.7 | **84.3** | **20.8** | 45.0 | **26.5** | **54.1** |

**Table 3: Performance comparison on zero-shot audio classification and tagging tasks. Models marked with ⁺/* are based on the HTS-AT/AST backbone.**

| Model | ESC-50 | US8K | VGGSound | FSD50K | AS |
|---|---|---|---|---|---|
| Cacophony | 93.4 | 77.1 | 27.0 | - | - |
| CompA | 89.1 | **85.7** | 29.5 | - | - |
| CLAP⁺ | 94.7 | 80.7 | 28.6 | 52.4 | 21.1 |
| MGA-CLAP⁺ | **94.9** | 83.7 | **31.8** | **54.5** | **23.0** |
| CLAP* | 91.6 | 76.6 | 26.8 | 47.8 | 16.9 |
| MGA-CLAP* | 92.0 | 79.4 | 29.2 | 49.7 | 19.3 |

of sound events, followed by PSDSm and PSDS2, which may pay more attention to remove confusion between classes [8].

## 5 RESULTS

### 5.1 Model Performance

*5.1.1 Performance on Coarse-grained Tasks.* We compare our proposed MGA-CLAP not only with the original CLAP but also with the SOTA model in each separate task. Specifically, for zero-shot retrieval, we involve FLAP (fusion) [57] and Cacophony [59] for comparison. The former is trained on LAION-Audio-630K using a more powerful audio encoder MAViL [16] and employs feature fusion proposed in [48] to process audios longer than 10s instead of directly cropping it. And the latter is trained on a 4M audio-text dataset with LLM re-captioning, which is much larger than our 450K pairs. While for zero-shot classification, we additionally involve CompA [12], which leverages an instruction-tuned Flan-T5-large model (770M) [34] as the text encoder and CompA-661K (an extension of LAION-Audio-630K) as the pre-training set.

The results concerning coarse-grained retrieval and classification tasks are reported in Table 2 and 3, respectively. As for retrieval,

**Table 4: Performance comparison on zero-shot sound event detection and text-to-audio grounding tasks. Models marked with [+]/[*] are based on the HTS-AT/AST backbone.**

| Model | DESED | | UrbanSED | | AS-S | TAG |
|---|---|---|---|---|---|---|
| | PSDS1 | PSDS2 | PSDS1 | PSDS2 | PSDS1 | PSDSm |
| UACA | 14.2 | 53.7 | 2.3 | 11.8 | 3.4 | 37.5 |
| WSTAG | 17.1 | 54.3 | 3.9 | 12.6 | 4.0 | 41.7 |
| PACL | 17.9 | 55.6 | 4.3 | 14.0 | 4.9 | 42.5 |
| CLAP[+] | 13.1 | 52.0 | 1.6 | 10.6 | 3.4 | 34.4 |
| MGA-CLAP[+] | **26.4** | **58.9** | **8.7** | **19.3** | 10.1 | 48.7 |
| CLAP[*] | 13.5 | 48.9 | 1.7 | 10.8 | 4.5 | 36.9 |
| MGA-CLAP[*] | 25.2 | 55.5 | 7.6 | 14.9 | **10.6** | **54.8** |

the proposed MGA-CLAP largely surpasses the original CLAP no matter which backbone is applied. When compared with current SOTA methods which involve more training resources, our MGA-CLAP is also competitive, achieving the best performance on 6 of 8 metrics. And for classification and tagging tasks, similar improvements over the original CLAP can also be observed. Noticeably, our MGA-CLAP with HTS-AT encoder reaches 31.8% accuracy on VGGSound, the most complex single-label classification dataset with 300+ classes, which is 2.3% higher than the previous SOTA, CompA. These above results underscore MGA-CLAP's ability to capture cross-modal alignment between texts and audio, leading to outstanding performance in versatile classification and retrieval tasks.

*5.1.2 Performance on Fine-grained Tasks.* We reimplement and retrain UACA [50] and WSTAG [54] using the CLAP paradigm since the original works only experiment on tiny datasets. Besides, we also reproduce PACL [31], a recent vision-language training framework, which employs cross-modal attention pooling to align local features with captions and demonstrates superior performance on fine-grained visual understanding tasks. All the above methods are implemented based on the HTS-AT backbone and we keep the training and evaluation settings consistent with MGA-CLAP.

As mentioned before, the original CLAP cannot uncover fine-grained alignment between frame features and text descriptions. As depicted in Table 4, it obtains extremely low scores, especially on time-sensitive metrics, such as PSDS1 and PSDSm. Although UCAC and WSTAG attempt to solve this problem, their results are still unpromising, since the frame-to-word interaction is modeled via simply score pooling. Besides, directly transferring PACL leads to a better but still suboptimal outcome. As a comparison, our MGA-CLAP with HTS-AT backbone obtains PSDS1 scores of 26.4%/8.7%/10.1% on DESED/UrbanSED/AudiosSet-Strong eval sets, which are 2x/5x/3x times those of the original CLAP. When switching the audio encoder to AST, better performances are witnessed on datasets containing more queries, such as AS-S and TAG.

## 5.2 Ablation Study

In this section, we ablate the module designs and hyper-parameter choices of the proposed MGA-CLAP. All experiments are conducted based on the HTS-AT backbone.

*5.2.1 Ablation Study on Each Sub-module.* Table 5 shows the model performance of the proposed MGA-CLAP trained with or without

**Table 5: Ablation study on sub-modules, where MC, LB, and HN denote modality-shared codebook, locality-aware block, and hard-negative guided loss, respectively. For retrieval and detection tasks, we solely list the R@1 and PSDS1 score.**

| MC | LB | HN | AC T2A | AC A2T | VGGSound | FSD50K | DESED | AS-S | TAG |
|---|---|---|---|---|---|---|---|---|---|
| | | | 39.7 | 51.9 | 28.6 | 52.4 | 13.1 | 3.4 | 34.4 |
| ✓ | | | 41.0 | 53.6 | 30.7 | 53.5 | 20.1 | 7.3 | 41.7 |
| | ✓ | | 39.4 | 51.8 | 28.5 | 52.9 | 21.2 | 5.6 | 41.1 |
| ✓ | ✓ | | 41.2 | 53.7 | 30.9 | 53.8 | 26.5 | 9.5 | 47.6 |
| ✓ | ✓ | ✓ | 41.8 | 54.4 | 31.8 | 54.5 | 26.4 | 10.1 | 48.7 |

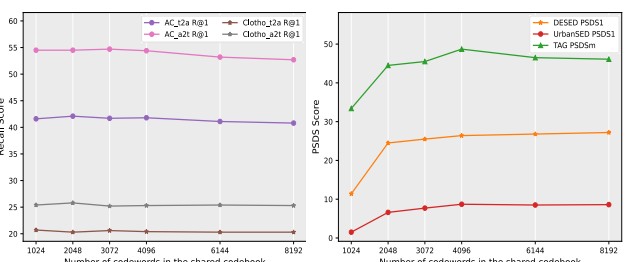

**Figure 4: Model performance on zero-shot retrieval (left) and detection and grounding (right) tasks with different numbers of codewords in the modality-shared codebook.**

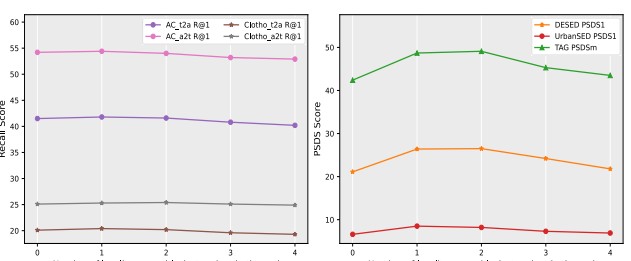

**Figure 5: Model performance on zero-shot retrieval (left) and detection and grounding (right) tasks with different numbers of locality-aware blocks.**

a specific sub-module. As seen, the incorporation of a modality-shared codebook boosts CLAP's understanding capabilities on both coarse-grained and fine-grained tasks, as it not only adopts common bases to represent global audio and text features but also links multi-modal local features with shared codewords. However, solely training with it cannot lead to satisfactory results on fine-grained tasks due to the inferiority of frame-wise representations. When further adopting the locality-aware encoder block, the PSDS scores on DESED, AS-S, and TAG datasets are improved by 5.1%, 2.2%, and 5.9%, respectively, suggesting the necessity of acquiring high-quality frame features. Additionally, simply involving the locality-aware block can also enhance the model performance on detection and grounding tasks. Finally, further equipped with the hard-negative loss, the whole system can achieve optimal results on each task as it can enhance the contrastive learning scheme.

*5.2.2 Ablation Study on the Size of Codebook.* We compare the modality-shared codebook in different sizes in Figure 5. As seen from the left figure, the R@1 scores on AudioCaps drop significantly

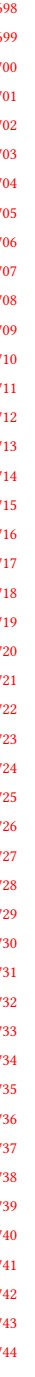

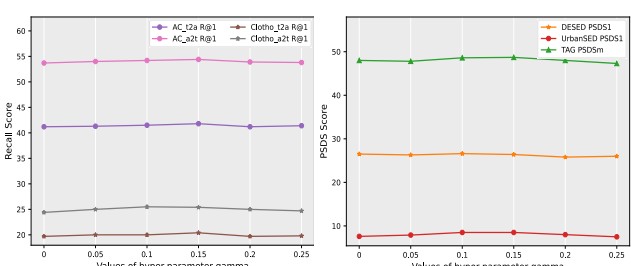

**Figure 6: Model performance on zero-shot retrieval (left) and detection and grounding (right) tasks with different values of hyper-parameter $\gamma$.**

when the number of codewords $M$ increases from 4096 to 8192. We argue that the enlarged codebook may bring about noisy activated codewords and irrelevant information while aggregating, making it difficult to retrieve matched pairs. Besides, it is also at risk of underfitting since some codewords may be undertrained. By contrast, although fewer number of codewords generally leads to slightly better outcomes on retrieval tasks, its performance on frame-level tasks decreases a lot as shown in the right figure. The possible reason is that each codeword must convey multiple semantics within a smaller codebook, thereby disturbing the frame-word interaction while seeking fine-grained alignment. Finally, we choose the number of codewords $M = 4096$ to make a trade-off between multi-grained tasks.

*5.2.3 Ablation Study on the Number of Locality-aware Block.* We conduct parameter analysis on the number of vanilla transformer blocks to be replaced by the locality-aware ones in Figure 5. It can be observed that the incorporation of locality-aware blocks contributes a lot to the enhanced capability due to the refinement of frame-wise features. Additionally, adopting 1 or 2 locality-aware blocks has similar effects on the downstream tasks. However, as the number grows, the performance degradation is witnessed. This is possibly due to more locality-aware blocks destroying the information flow and pre-trained knowledge in the transformer backbone.

*5.2.4 Ablation Study on the Values of $\gamma$.* As stated before, $\gamma$ in Equation (8) controls the difficulty of negative samples with a higher value paying more attention to harder ones. We then study the effects of its numerical values in Figure 6. The results indicate that $\gamma = 0.15$ or $\gamma = 0.10$ generally yields better outcomes as a larger one may overemphasize the hard negative samples and potentially neglect the relation with other in-batch data points.

*5.2.5 Ablation Study on the Design Choice of Codebook.* We ablate the detailed designs, namely the max pooling to compute the affinity scores and the Sparsemax function to normalize aggregation weights, in the modality shared codebook and provide the related outcomes in Table 6. As shown, if applying the mean pooling instead of max pooling, some non-salient local cues may be overwhelmed by the primary sound. Then severe performance drops can be found in tagging, detection and grounding tasks, where local patterns play an important role in recognition. And when changing the activation function to Softmax, the system produces poor results on all tasks, which is only slightly better than the original CLAP. We

**Table 6: Ablation study on designs of the codebook, where -, (1), (2) denote the current setting, replacing max pooling with mean pooling, replacing Sparsemax with Softmax.**

| Design | AC T2A | AC A2T | VGGSound | FSD50K | DESED | AS-S | TAG |
|--------|--------|--------|----------|--------|-------|------|------|
| -      | 41.8   | 54.4   | 31.8     | 54.5   | 26.4  | 10.1 | 48.7 |
| (1)    | 40.9   | 52.9   | 30.2     | 52.7   | 15.3  | 5.6  | 39.0 |
| (2)    | 40.2   | 52.3   | 28.5     | 52.6   | 13.4  | 4.1  | 35.8 |

| Codeword Id | Codeword to Phrase Similarity (top 3) | Codeword to Frame Similarity |
|-------------|----------------------------------------|------------------------------|
| # 2917 | Bark: 0.246 
 Yip: 0.219 
 Dog: 0.219 | Dog: [0.1, 7.7], [9.0, 10.0] — Dog: [0.7, 2.7], [4.0, 6.3], [6.8, 9.5] |
| # 3730 | Female speech: 0.185 
 Conversation: 0.131 
 Speech: 0.126 | Female speech: [2.8, 4.6], [5.0, 6.1] — Female speech: [1.8, 2.6], [3.2, 3.9], [8.0, 9.2] |
| # 3830 | Male speech: 0.178 
 Conversation: 0.168 
 Speech: 0.168 | Male speech: [8.4, 9.7] — Male speech: [0.0, 2.7] |
| # 3678 | Sewing machine: 0.188 
 Pulleys: 0.179 
 Lawn mower: 0.175 | Lawn mower: [1.0, 6.2] — Sewing machine: [0.0, 3.4], [7.5, 10.0], Female speech: [0.0, 1.3], [1.9, 8.7] |

Ground truth sound event boundary
Codeword to frame feature similarity

**Figure 7: Visualization of codewords' role in connecting text modality (column two) and audio modality (column three). Notice that the third column shows the ground truth sound events boundary (dotted box) and the frame-level similarity with the codeword (solid line) of two example audios.**

argue that with Softmax, the aggregation weights are no longer sparse, introducing many noisy components when representing the global features. As a result, the semantic meanings of codewords may be blurred and the connection between frame and word will be influenced.

## 5.3 Visualizations

*5.3.1 Semantics of Codewords.* In this subsection, we try to disclose the meanings of some representative modality-shared codewords. For a specific codeword, we first compute its cosine similarity with textual features of sound classes taken from AudioSet taxonomy to find out its semantics. Then we compute its cosine similarity with frame representations to examine if it also correlates with acoustic features. The results are given in Figure 7. Taking the 1st row as an example, the 2917th codeword has a large similarity with textual descriptions related to "dog", suggesting its represented semantics. Besides, as shown in the two sub-figures, the cosine similarity between the codeword and frame features also shows synchronization with the temporal location of sound events. Specifically, the scores are high when a dog bark is truly presented while low when the sound is absent. When comparing the 2nd and 3rd rows, it can be seen that the codeword can encode finer acoustic attributes, such

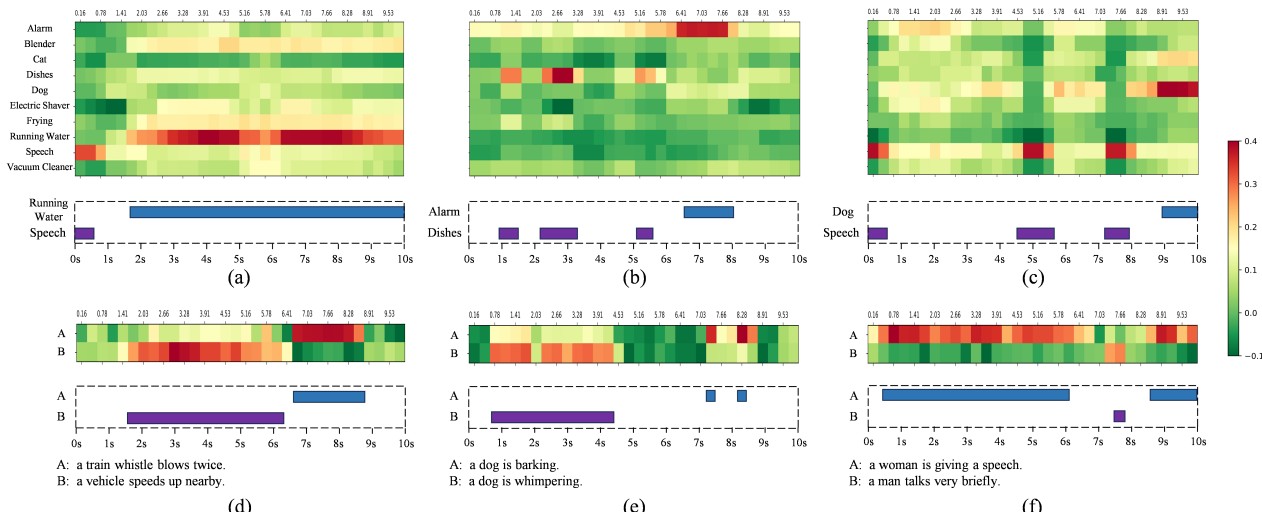

Figure 8: Successful examples of achieved fine-grained alignment. For each sub-figure, the frame-level similarity with textual features of sound classes or detailed captions and the temporal location of sound events are visually depicted.

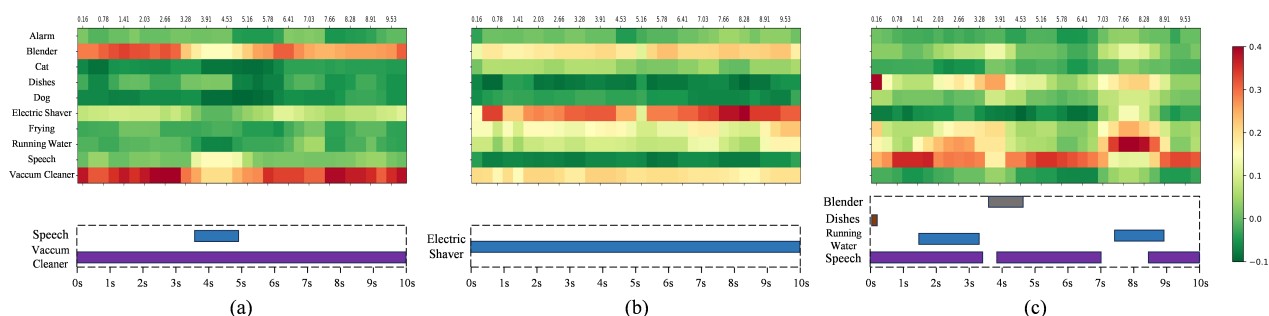

Figure 9: Failure examples of fine-grained alignment detached from MGA-CLAP.

as the gender of speakers. Finally, as for the 4th row, the semantics of rare sound classes (e.g., sewing machine) can also be learned from the MGA-CLAP pipeline. And from the last figure, one can see that the semantic mapping is salient even under polyphonic environments.

### 5.3.2 Fine-grained Alignment.
We provide several examples of MGA-CLAP achieving fine-grained alignment in Figure 8. The cases show that our method may capture both frame-to-phrase (seen from the first row) and frame-to-caption (seen from the second row) correspondence, thereby obtaining promising results on zero-shot detection and grounding tasks. Surprisingly, it can tell apart barking and whimpering sounds at frame-level as seen in sub-figure (e), which are both made by dogs but varied in pitches, suggesting that the subtle semantics of captions are also well aligned with acoustic characteristics. Moreover, we also visualize some bad cases in Figure 9. Currently, our MGA-CLAP may be confused about similar sounds such as blender and vacuum cleaner (seen from Figure 9 (a)) and fail to capture long-duration dependency sometimes (seen from Figure 9 (b)). Moreover, it may omit certain sounds (such as

blender in Figure 9 (c)) especially when multiple acoustic events take place simultaneously.

## 6 CONCLUSION

In this work, we devise MGA-CLAP to align audio features with language descriptions from both coarse-grained and fine-grained views. To achieve this goal, MGA-CLAP employs a codebook to construct a shared feature space for cross-modal interaction and optimize its internal codewords carefully to seek frame-word correspondence. Based on the modality-shared codebook, a novel encoder block is designed to enhance the salience of local patterns while a re-weighting loss term is considered to mine hard-negative pairs during optimization for better cross-modal alignment. By pre-training on large audio-text datasets, our MGA-CLAP not only outperforms the baseline CLAP to a large extent but also yields better or competitive outcomes on versatile language-audio understanding tasks when compared with current SOTA variants. Through extensive visualizations and ablation studies, the effectiveness of proposed designs is verified.

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
