# OpenReview forum: "Advancing Multi-grained Alignment for Contrastive Language-Audio Pre-training"
_acmmm.org/ACMMM/2024/Conference — MM2024 Oral_

### Official Review · Reviewer_UauH · 2024-05-14

**Rating:** 6
**Confidence:** 3

**Summary:**

This paper proposes MGA-CLAP, where audio features and language descriptions are aligned from both coarse-grained and fine-grained viewpoints. The authors adopt a modality-shared codebook to encourage the multi-modal features to interact on a finer granularity. Their experimental results in various tasks demonstrate the effectiveness of the proposed framework. In addition, extensive ablation studies are conducted to show the effects of each component.

**Strengths:**

- Dealing with an important topic regarding audio-text cross-modal modeling.
- Well-designed training framework based on a deep insight into the characteristics of audio-text paired data. Each sound event inherently happens in a temporarily local region, whereas most contrastive representation learning frameworks do not consider this property of sound events. The unawareness of this may deteriorate the performance of cross-modal representation. This study addresses this issue.
- Comprehensive experiments from audio-text retrieval to zero-shot audio classification, audio tagging, sound event detection, and text-to-audio grounding tasks. The proposed MGA-CLAP performs very well on all conducted tasks. Ablation studies are also sufficient, which answer questions most readers will have.

**Limitations:**

- I'm wondering if this model is easily trained or not. More specifically, does introducing a codebook/locality-aware block affect convergence speed or training stability?
- The authors can contribute more to the community by making the code and pretrained checkpoint publicly available
- I'm interested in whether trained text features can improve the performance of text-to-audio generation models (such as AudioLDM). This is a minor comment from my curiosity, which I do not take into account in rating the paper because this is out of the scope.

**Suitability:**

3

---

### Official Review · Reviewer_jeMY · 2024-05-20

**Rating:** 4
**Confidence:** 4

**Summary:**

This paper proposes a framework to improve both coarse- and fine-grained audio-language alignment in large-scale contrastive pre-training. Specifically, the paper employs a codebook to construct a shared feature space for cross-modal interaction and optimize its internal codewords carefully to seek frame-word correspondence. And then a novel encoder block is designed to enhance the salience of local patterns as a hard-negative guided loss is devised to boost alignment effects.

**Strengths:**

The method proposed in the paper is novel and interesting, utilizing a shared codebook to represent multi-modal global features with common bases. This approach could be beneficial for some multi-modal tasks, such as multi-modal alignment and explainability. Additionally, the paper conducts extensive experiments and provides thorough ablation studies on different modules of the model. The experimental results are well visualized. In the end I believe that this work is a good contribution to the community.

**Limitations:**

In Section 4.2 of the paper, it is mentioned that "Note that for each specific task, we would pre-train a new model from scratch with a newly constructed dataset ...", does the newly constructed dataset here refer to the dataset mentioned in Section 4.1 and the task-specific datasets in Table 2 combined? And why not directly merge and use the training sets of all tasks to train a powerful CLAP model?
Additionally, there are some minor errors in the symbols used in the paper, please check and correct them, for example, line 286.

**Suitability:**

3

---

### Official Review · Reviewer_wgvs · 2024-05-23

**Rating:** 4
**Confidence:** 3

**Summary:**

The paper presents an advanced method for improving multi-grained alignment in contrastive language-audio pre-training by introducing a modality-shared codebook and a locality-aware block. This approach enhances both coarse-grained and fine-grained alignment, addressing the limitations of existing models like CLAP.

**Strengths:**

(1) By concentrating on local patterns, the locality-aware block enhances the ability to capture detailed audio events and their corresponding text descriptions. This improvement boosts performance in tasks that demand precise timing information, like sound event detection and text-to-audio grounding.

(2) Authors conducted  extensive experiments on various datasetes for zero-shot coarse- and fine-grained evaluation  on different tasks, demonstrating the model's superior performance compared to both the baseline CLAP and other state-of-the-art methods.

(3) This approach of using a shared codebook for aligning multimodal features is relatively noval and addresses the granularity gap between audio frames and text tokens.

(4) Enhancing contrastive learning with hard negatives is a known technique, but its specific application and adaptation for multimodal audio-text alignment provide a unique contribution.

**Limitations:**

(1) Comparative Analysis with other state of the art papers: Comparative analysis with other state of the arts methods such as such as AudioCLIP [1], Wav2CLIP [2], BEATs [3], and PANNs [4] is  limited. Comparison with these method  would highlight MGA-CLAP’s unique advantages.

(2) My doubt is by removing the q-k attention mechanism, the model loses the ability to incorporate global context. This might lead to fragmented representations, where each frame is considered in isolation, potentially missing out on relationships between frames that span longer durations.

(3) The reliance on Sparsemax for normalizing aggregation can lead to codeword sparsity that could potentially lead to underutilization of available codewords. This might result in suboptimal performance when dealing with highly diverse audio and text data, where more nuanced representations could be beneficial.

(4) To determine the effectiveness of fine-grained features it will be interesting to see how the model performs on generating captions for given audio clips and capturing detailed semantic information.

(5) How current architecture if effective against noise.?The current evaluation of the MGA-CLAP model lacks an analysis of its robustness to noise. Incorporating noise robustness experiments will provide a comprehensive understanding of the model performance, especially for fine-grainded audio tasks.



[1] Guzhov, Andrey, Federico Raue, Jörn Hees, and Andreas Dengel. "Audioclip: Extending clip to image, text and audio." In ICASSP 2022-2022 IEEE International Conference on Acoustics, Speech and Signal Processing (ICASSP), pp. 976-980. IEEE, 2022.

[2] Wu, Ho-Hsiang, Prem Seetharaman, Kundan Kumar, and Juan Pablo Bello. "Wav2clip: Learning robust audio representations from clip." In ICASSP 2022-2022 IEEE International Conference on Acoustics, Speech and Signal Processing (ICASSP), pp. 4563-4567. IEEE, 2022.

[3] Chen, Sanyuan, Yu Wu, Chengyi Wang, Shujie Liu, Daniel Tompkins, Zhuo Chen, and Furu Wei. "Beats: Audio pre-training with acoustic tokenizers." arXiv preprint arXiv:2212.09058 (2022).

[4] Kong, Qiuqiang, Yin Cao, Turab Iqbal, Yuxuan Wang, Wenwu Wang, and Mark D. Plumbley. "Panns: Large-scale pretrained audio neural networks for audio pattern recognition." IEEE/ACM Transactions on Audio, Speech, and Language Processing 28 (2020): 2880-2894.

**Suitability:**

3

---

### Official Review · Reviewer_1x5C · 2024-05-23

**Rating:** 5
**Confidence:** 3

**Summary:**

This paper introduces a novel approach for improving both coarse- and fine-grained audio-language alignment in large-scale contrastive pre-training. The method involves using a shared codebook to unify the granularity and latent distribution of audio and text features. A locality-aware block purifies local patterns, and a hard-negative guided loss boosts alignment effects.  These paper gives the following contributions:
1. Contrastive Language-Audio Pre-training (CLAP): Previous models aggregate uni-modal local representations into global ones for coarse-grained alignment but often ignore fine-grained text-audio challenges.
2. Shared Codebook: This approach uses a shared codebook to represent multi-modal global features with common bases, encoding modality-shared semantics.
3. Locality-aware Block: A new block purifies local patterns to enhance frame-level features.
4. Hard-Negative Guided Loss: This loss function emphasizes hard negative samples to improve alignment.

**Strengths:**

1. Good Approach: The introduction of a shared codebook for unifying granularity and latent distribution between audio and text features is a novel contribution.
2. Extensive Evaluation: The model is tested on eleven different zero-shot coarse- and fine-grained tasks, showcasing its versatility and robustness.
3. Significant Improvements: MGA-CLAP shows marked improvements over baseline and competitive results against SOTA models.

**Limitations:**

The proposed method is interesting, and the experiments also support the claims.  One of conerns is about the Comparisons method, author should better present the baselines's parameters and training data size in the Table. So reader can better understand the advantiges.

**Suitability:**

3

---

### Meta-Review · Area_Chair_9paa · 2024-07-01

**Recommendation:** Accept (Oral)
**Confidence:** 4

**Metareview:**

The paper introduces MGA-CLAP, which incorporates several small extensions over the original CLAP, where audio features and language descriptions can be aligned from both coarse-grained and fine-grained viewpoints. Authors introduce an additional small codebook which is shared across modalities to encourage the multi-modal features to interact on a finer granularity. Experimental results on various tasks demonstrate the effectiveness of the proposed framework, setting SOTA on some tasks. Reviewers asked for additional comparisons and clarifications which authors were able to provide. While the paper does not have fundamental novelty, it presents a strong contribution to cross- and multi-modal representation learning and is thus a good candidate for ACM MM.

Reasons to accept:
- The paper improves on a fundamental and important approach in audio/ text modeling.
- The modeling framework is well designed and supports temporal localization of sound events, which adds to the quality and novelty of the work over other contrastive frameworks.
- Authors conduct comprehensive experiments from audio-text retrieval to zero-shot audio classification, audio tagging, sound event detection, and text-to-audio grounding tasks, showing the good performance of the proposed approach. In the rebuttal, authors provided additional baselines on these tasks.
- Authors confirm code and checkpoints will be released.

Reasons to reject:
- Reviewers still have a number of questions that would deserve more detailed answers, e.g. the consequences of using sparsemax, noise robustness, and trade-off between local and global features.